# Electric fields control the orientation of peptides irreversibly immobilized on radical-functionalized surfaces

Lewis J. Martin [1], Behnam Akhavan [1,2] & Marcela M.M. Bilek [1,2,3,4]

Surface functionalization of an implantable device with bioactive molecules can overcome adverse biological responses by promoting specific local tissue integration. Bioactive peptides have advantages over larger protein molecules due to their robustness and sterilizability. Their relatively small size presents opportunities to control the peptide orientation on approach to a surface to achieve favourable presentation of bioactive motifs. Here we demonstrate control of the orientation of surface-bound peptides by tuning electric fields at the surface during immobilization. Guided by computational simulations, a peptide with a linear conformation in solution is designed. Electric fields are used to control the peptide approach towards a radical-functionalized surface. Spontaneous, irreversible immobilization is achieved when the peptide makes contact with the surface. Our findings show that control of both peptide orientation and surface concentration is achieved simply by varying the solution pH or by applying an electric field as delivered by a small battery.

[1] School of Physics, University of Sydney, Sydney, NSW 2006, Australia. [2] School of Aerospace, Mechanical and Mechatronic Engineering, University of Sydney, Sydney, NSW 2006, Australia. [3] Charles Perkins Centre, University of Sydney, Sydney, NSW 2006, Australia. [4] University of Sydney Nano Institute, University of Sydney, Sydney, NSW 2006, Australia. Lewis J. Martin and Behnam Akhavan contributed equally to this work. Correspondence and requests for materials should be addressed to B.A. (email: behnam.akhavan@sydney.edu.au) or to M.M.M.B. (email: marcela.bilek@sydney.edu.au)

Modern medicine increasingly relies on implantable biomedical devices[1]. The function(s) of these devices are often limited due to unsuccessful integration with host tissue; and in extreme cases this necessitates replacement of the device by revision surgery[2]. Bio-devices such as pacemakers and bone implants can cause unfavourable reactions in the surrounding host tissue, e.g. foreign body responses that lead to encapsulation by fibrotic tissue or the formation of bacterial biofilms resulting in untreatable infections[3]. These responses cause significant pain and suffering, as well as considerable economic burden for national health-care systems[4]. Functional coatings can mitigate such problems by masking the implanted devices and promoting successful integration with the body. Biologically functionalized surfaces, in particular, have the potential to direct optimal host responses by providing biological cues through molecules immobilized at the host-device interface[5].

Bio-functionalized devices are surface-engineered to present bioactive molecules. These molecules influence the biology of nearby cells by providing signals via specific interactions with cell surface receptor proteins[6]. Covalent bio-functionalization is necessary to avoid adsorption-induced denaturing[7] and/or loss of the biomolecules through protein exchange such as occurs in the Vroman effect[8,9]. Established methods for covalent attachment, however, involve cumbersome multistep wet chemistry, often using reagents that may present hurdles for regulatory approval[10]. Recently, a nonspecific, chemical linker-free approach to achieve covalent attachment of bioactive molecules directly from buffered solution has been demonstrated[11]. Covalent immobilization is achieved through reactions with radicals embedded under the surface by energetic ion bombardment[12]. Radical-functionalized surfaces can be created on both non-polymeric[13,14] and carbon-rich polymeric surfaces[15,16]. The buried radicals are capable of diffusing to the surface[17], where they react with biomolecules. The ion-treated surfaces are typically hydrophilic due to reactions with atmospheric oxygen[18]; thus, the immobilized proteins do not suffer denaturation through physical interactions with the surface. Due to the non-specific nature of the reaction, immobilized orientation could be controlled by orienting the biomolecules on approach.

Protein-functionalization of surfaces has been applied to improve the biocompatibility of medical devices[6]. Practical applications are, however, impeded by loss of function in post-packaging sterilization, denaturation-induced thrombosis[19] and the possibility of pathogen transfer from proteins produced in micro-organisms[20]. Surface functionalization with bioactive peptides may offer a solution. The smaller size of peptides compared to proteins makes them more resilient to sterilization. Furthermore, peptides can be generated synthetically as opposed to in microorganisms. Mimetic peptides are derived from the active amino acid sequences of proteins and therefore can provide the key functionality of the protein[21]. The most commonly used peptide mimics are derived from extracellular matrix proteins such as fibronectin or collagen, since they support native cell adhesion and hence integration of an implanted surface into tissue[22,23]. The RGD peptide, for example, has often been used as a proxy for fibronectin[24,25], and there are a number of other peptide mimics of growth factors[26,27].

Surface-attached biomolecules can interact with the micro-environment and direct cell behaviour only if bound in an orientation that allows access to the active site[28,29] and in a conformation in which the active site is structurally intact[30,31]. The density of surface attachment also plays a role in determining the effectiveness of the immobilized biomolecules especially with reference to promoting adhesion and spreading of cells[32]. The cell adhesive activity of the RGD peptide, in particular, is known to be sensitive to surface concentration[33], and molecular crowding of immobilized peptides has been shown to affect presentation of the active site[34,35].

Control of the surface orientation of peptides by changing their chemistry and achieving site-specific attachment has been studied before. For example, site-specific phosphorylation of serine residues in a silicon-based peptide[36] or the insertion of a cysteine residue[37]. However, controlling the orientation of peptides with these traditional chemical approaches is typically cumbersome and tedious for large-scale manufacturing. Electric fields might present an opportunity to orient peptides at biomaterial interfaces. Charged interfaces repel and attract like-charged and oppositely charged regions of a biomolecule, respectively, resulting in orientation of molecules that are asymmetrically charged.

The electric field at the surface can be manipulated by changing the solution pH or by the application of an external electric field. Electric field-induced bidirectional alignment along the long axis of large proteins has been achieved with DC fields[38] and in dielectrophoresis with AC electric fields[39]. A unidirectional orientation can only be achieved via a permanent dipole moment and hence requires small molecules ($<\sim$100 kDa), for which the induced dipole moment does not dominate[40]. Furthermore, as the dielectrophoretic force scales with particle size, increasing the arrival rate and surface immobilized density of peptides would require unfeasibly high field strengths to overcome thermal motion[41].

Here we demonstrate the use of electrostatic manipulation to control the orientation of peptides through electric field interactions with permanent dipoles. Guided by computational simulations, we strategically design a peptide incorporating a FLAG-tag functional epitope used in generating fusion proteins. A radical-functionalized plasma polymer (RFPP) surface, with a high concentration of reactive radicals, is used to covalently immobilize this peptide upon arrival at the surface. We demonstrate the control of both concentration and orientation of the immobilized peptide by varying the solution pH initially and then extend the achievable range by employing externally applied electric fields. Our findings shed light on mechanisms of biomolecule immobilization that are extremely important for the design of synthetic peptides and the production of advanced bio-functionalized materials.

## Results

**Radical-functionalized surface.** We tested the ability of electric fields to control the orientation of peptides as they approach and bind to a radical-functionalized surface (Fig. 1). To do this, we used an RFPP coating. The coating was fabricated on titanium substrates using a unique plasma polymerization configuration, where the substrate is negatively biased in a pulsed manner during the deposition. Pulsed biasing of the substrate results in enhanced bombardment of the growing film by accelerated ions, thus allowing the generation of a high concentration of radicals within the structure of the coating. The EPR spectrum of the RFPP coating confirms that the surface is permeated by radicals (Fig. 2a). The EPR spectrum shows a single resonance peak, centred at 3513G (g-value of 2.003), ascribed to unpaired electrons associated with radical-containing compounds. The radicals are capable of forming covalent bonds with biomolecules, as previously demonstrated on similar plasma polymerized structures[42,43]. Plasma polymerization parameters were optimized to produce mechanically and chemically robust RFPP coatings capable of covalently binding peptide molecules directly from solution at a wide range of pH values.

Surface chemistry and charge are the most important characteristics influencing the interaction of surfaces with peptides[44,45]. The XPS survey spectrum of RFPP coating shows

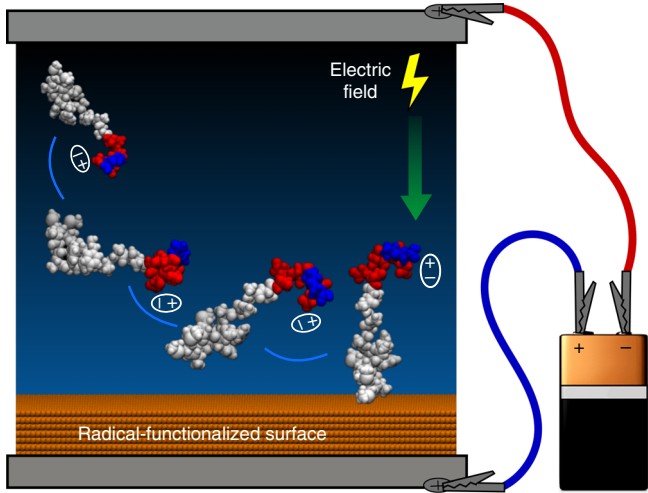

**Fig. 1** Control of peptide orientation by electric field. Charge separation on one end of the peptide creates a dipole moment (indicated by ellipses) that aligns with the electric field and rotates the entire molecule. Once contact is established with the radical-functionalized surface, covalent linkage anchors the peptide in this orientation

that the surface is composed of carbon, nitrogen and oxygen with atomic concentrations of 71.3, 19.6 and 9.1%, respectively (Fig. 2b). No titanium signal is detected, indicative of complete coverage of the underlying titanium surface by the RFPP layer with a thickness of 32.6 nm as measured by spectroscopic ellipsometry. The detected oxygen is primarily originating from post-deposition oxidation, an inevitable process that occurs upon the exposure of samples to atmospheric oxygen[46,47].

The high resolution C 1 s XPS spectrum was fitted with components associated with C–C/C–H at binding energy (BE) ≅ 284.6 eV, C–O/C–N at BE ≅ 286.5 eV, C=O/N–C=O at BE ≅ 287.5 eV and COOH at BE ≅ 289 eV (Fig. 2d)[48,49]. The four components indicate the broad range of chemical environments within the RFPP, in particular C=O and COOH moieties that become charged in aqueous environments. The presence of nitrogen and oxygen on the surface suggests that both basic and acidic groups are present, leading to the formation of charged moieties such as $NH_3^+$ and $COO^-$. The presence of these compounds allows modulation of the surface charge by altering the immobilization solution pH. The changes of surface zeta potential as a function of pH indicate that the isoelectric point (where the surface has no charge) is at pH = 4.5 (Fig. 2e). The surface becomes progressively more positive below this point and progressively more negative above it. This behaviour allows us to determine the effect of surface charge on interfacial interactions with a charged peptide that has a constant surface charge over this pH range.

**Peptide design and simulation.** The FLAG peptide was designed in such a way as to allow the evaluation of the influence of electric fields on the orientation and concentration of the immobilized peptide. The sequence (Ac-FFMMMAAAAAAAAAADDDDDK-NH$_2$) demonstrates several features that facilitate this evaluation: Methionine (M) residues introduce sulfur so that surface concentration can be measured by XPS. These residues are proximate to phenylalanine (F) residues that form a hydrophobic region (FFMMM) on one side of the peptide. On the other side of the peptide, there is a sequence based on the FLAG epitope[50] (DDDDK) capable of binding antibody molecules. This segment allows for the sensitive ELISA assay to assess how the surface immobilized peptide is oriented. This sequence also introduces

both a net negative charge of −4 e and an electric dipole created by the aspartic acid (D) and lysine (K) residues. Crucially, due to the close proximity of the charges in the sequence, they cannot be separated and the dipole destroyed by structural changes in the peptide. Additionally, the peptide is unlikely to fold due to the different polarities of the hydrophobic and hydrophilic ends. The separation of these ends is supported by a bridge of alanines (A), which can exhibit a disordered structure. This arrangement allows the peptide to be linear, with definitive orientations with respect to the surface, one of which provides access to the charged epitope.

To predict the solution structure of the peptide, we performed an 800 ns equilibrium simulation in water. The secondary structure[51] shows a clear majority population consisting of a beta-turn through residues 3–7 connected by a beta-bridge between residues 2 and 10, while other residues are in random coil conformation (Fig. 3a). While there are minor populations, defined mainly by a beta-turn in one or two of residues 8–21, these do not correspond to folding of the peptide and do not change the accessibility of the FLAG epitope. This is demonstrated by the time series of the radius of gyration ($R_g$) (Fig. 3b). It is clear that after 40–50 ns, the peptide is equilibrated from its fully extended starting structure. The initial reduction in $R_g$ from full extension corresponds to folding into a beta-turn within the hydrophobic FFMMM- region. After this point, fluctuations to minor populations that include beta-turns do not significantly affect the peptide extension.

The charged region of the peptide creates a dipole moment that may affect the orientation of the peptide in an electric field. The vector angle between the dipole moment and the peptide principal, or 'long', axis fluctuates around 12.5 degrees, indicating close alignment between the dipole and the peptide direction of extension (Fig. 3c). This ensures that in the presence of an electric field, when alignment of the dipole is achieved the whole peptide will become parallel to the electric field lines.

The residues in the charged region, i.e. lysine (K) and aspartic acid (D), hold different charge states depending on the pH of the solution. Changing the pH alters the peptide net charge, which also alters the interaction with charged surfaces that controls the arrival rate and surface concentration (Fig. 3d). Changing the pH can thus remove or alter the strength of the dipole, affecting the peptide orientation at the surface. When the charged region is maximally charged, the peptide carries a net charge of −4 e. This situation occurs within the pH range of 5–10, where all residues are at least 90% charged. The peptide immobilization was therefore carried out at two pH values of 5.4 and 9.8, striking a balance between low and high RFPP surface charge, respectively, and optimal charging of the peptide.

A representative peptide structure chosen from the simulation trajectory demonstrates the peptide features (Fig. 3e). The beta-turns across the hydrophobic residues (F, M) lead to a folded section, likely formed in order to reduce water contact. The less-hydrophobic alanine (A), as per our design, forms a linear bridge between this region and the charged, polar FLAG epitope with an electric dipole aligned along the peptide principal axis. According to these simulation results, the designed sequence successfully constructs a linear, unstructured peptide that exposes the FLAG epitope to solution.

**Orientation and concentration control by pH.** XPS and ELISA can be used to detect the presence of surface protein or peptide. Here, we use XPS to measure the concentration of surface-bound peptide via the sulfur atomic concentration from methionine. The absence of sulfur signals in the XPS data of the RFPP coating before immobilization (<0.15 at. %, the XPS signal sensitivity) allows the use of this measurement to quantify the surface-bound

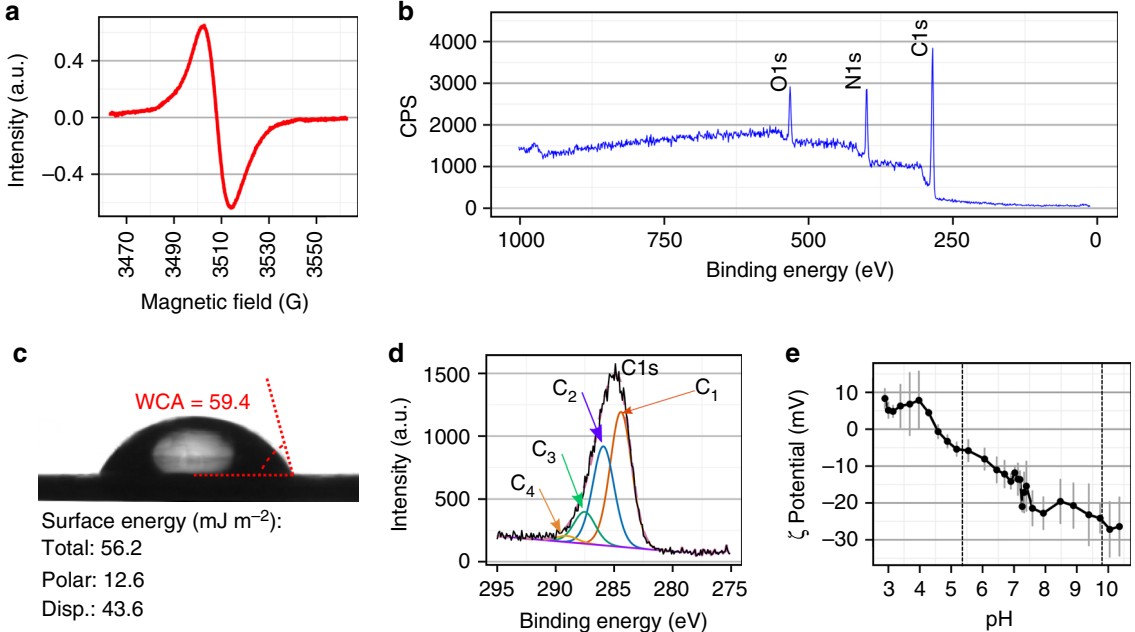

**Fig. 2** The surface has radicals and a range of charge states. **a** Electron paramagnetic resonance spectrum of the radical-functionalized plasma polymer (RFPP) showing a broad and symmetrical peak, indicative of unpaired electrons associated with radicals. **b** X-ray photoelectron spectroscopy (XPS) survey spectrum of the RFPP film showing a surface elemental composition of carbon (71.3 at.%), nitrogen (19.6 at.%) and oxygen (9.1 at.%). No sulfur or titanium is detected on the surface. **c** Water contact angle (WCA) of the RFPP surface. The total, polar and dispersive (Disp.) surface energies are given. **d** XPS C 1 s high-resolution spectra of the RFPP fitted with four components, $C_1$: C−C/C−H, $C_2$: C−O/C−N, $C_3$: C=O/N−C=O and $C_4$: COOH. **e** The changes of zeta potential as a function of pH show that the surface is negatively charged above pH 4.5, with the negative charge saturating at pH $\cong$ 7.5. The dashed lines indicate the pH values used in the immobilizing solutions. Error bars are s.d. (**e**)

peptide. We also use ELISA to indicate the presence of the functional FLAG epitope. For this experiment, the peptide was immobilized at pH 5.4, the lower of the two pH values used, because the surface charge is nearly neutral at this value.

The XPS and ELISA detection of surface-bound peptide increase with increasing concentration of peptide in the immobilization solution (Fig. 4a, b). The sulfur atomic concentration is undetectable below 5 µg mL$^{-1}$ (Fig. 4a), but the increase after this point indicates an increasing density of surface immobilized peptide. The ELISA assay indicates that the FLAG epitope is accessible by antibody in solution, and that this assay is concentration-dependent up to 20 µg mL$^{-1}$ (Fig. 4b). At concentrations higher than this, the signal plateaus and saturates (see Supplementary Fig. 1), most likely because steric hindrance, due to the large size (relative to the peptides) of the antibodies, prevents further antibody binding. This proof-of-concept experiment illustrates that the density of the immobilized peptide is measurable over a wide range of surface immobilized concentrations by a combination of XPS and ELISA. While the sensitivity of XPS is limited to immobilizing concentrations above 5 µg mL$^{-1}$, the ELISA assay is sensitive to peptide concentrations at the lowest levels tested, but the signal saturates at higher concentrations. The combined use of XPS and ELISA can, therefore, measure the presence of peptide on the surface at both high and low surface densities.

The ELISA assay should also be capable of differentiating the orientations of bound peptide. Here we assign FLAG$^{UP}$ to peptide bound through the hydrophobic end, which presents the FLAG epitope into solution, and FLAG$_{DOWN}$ to peptide bound via the FLAG epitope itself. It has been previously reported that surface-immobilized peptides can lie horizontally as opposed to normal to the surface[34,52,53], potentially concealing the FLAG epitope even in the FLAG$^{UP}$ orientation. This situation is, however, unlikely to happen for our study since the epitope is charged and

has lower energy when fully hydrated as compared to that when contacting the RFPP surface. In addition, the presence of the alanine spacer between the hydrophobic end and the FLAG epitope should increase antibody binding in the FLAG$^{UP}$ orientation by facilitating more contact in solution, a phenomenon known as the spacer effect[25,34,54].

The solution pH was changed during the immobilization to explore the effect of surface charge on peptide attachment. Increases in pH increase the negative surface charge and demonstrate the effect of the associated electric field. Informed by the zeta potential measurements (Fig. 2e), we chose immobilizing solution pH values of 5.4 and 9.8 to compare peptide binding while remaining within the constraints of our buffer system. For this experiment, the peptide concentration during immobilization was chosen to allow detection by XPS, and hence was in the saturated region of concentrations for ELISA.

The XPS analysis of peptide immobilized at the pH 9.8 condition yielded a zero sulfur signal, while ~ 0.7% sulfur was detected for the sample prepared at pH = 5.4 (Fig. 4c). The peptide maintains a constant charge of −4 e across both pH conditions (Fig. 3d). It is, therefore, the surface charge only that controls the arrival rate and binding of the peptide to the RFPP film. The RFPP surface is several-fold more negatively charged at pH 9.8 compared to pH 5.4, and hence the surface gives rise to an electric field that repels negative charges. The RFPP surface at such conditions represents a potential barrier that results in a slower peptide arrival rate to the surface and consequently a lower degree of covalent binding over the incubation time. Even though the sulfur level is not detectable by XPS at pH 9.8, there may be peptide bound on the surface. For the pH = 5.4 condition, where sulfur signals are detected, the surface is nearly neutral and so does not have a steep potential barrier for peptide binding. This surface-binding behaviour is consistent with the trends observed for adhesion of proteins on charged surfaces[55–58].

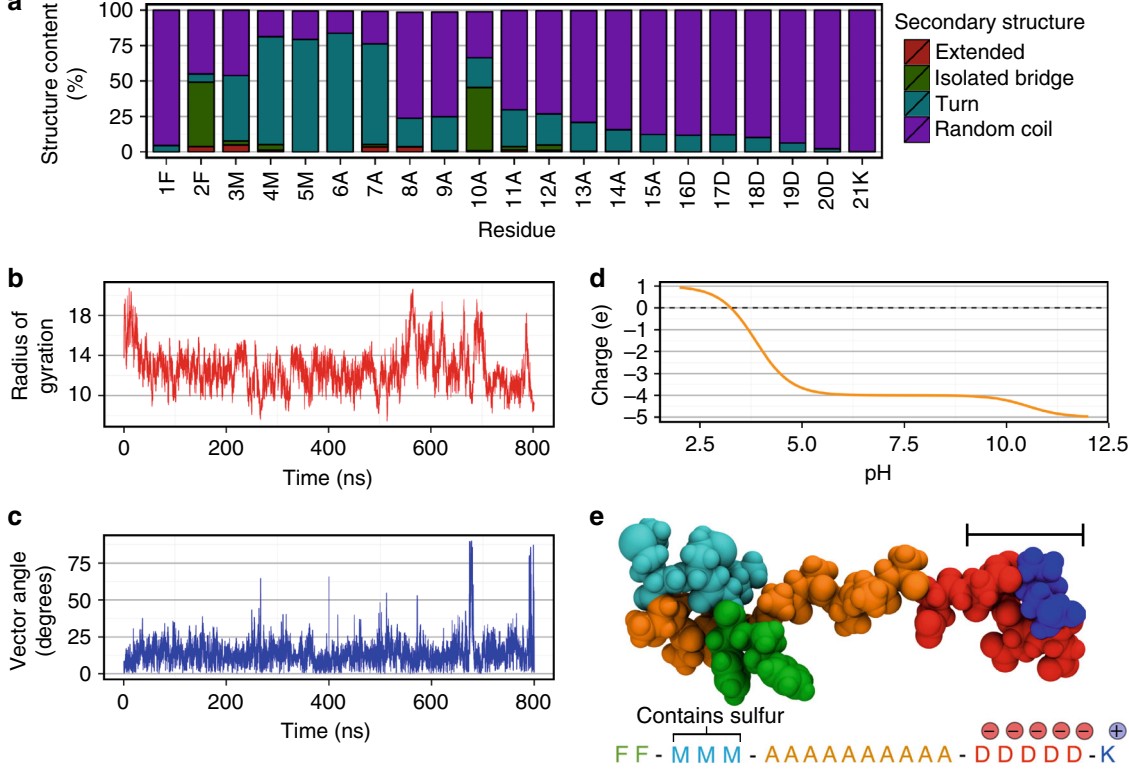

**Fig. 3** The simulated peptide is linear and mostly unstructured. **a** Stacked bar chart labelled with amino acid one-letter codes showing the per-residue structure content that is mostly random coil. **b** The time series of $R_g$ shows that after 40 ns of equilibration, the peptide maintains a single length. **c** Time series of the vector angle between the first principal axis and the dipole moment created by the charged residues. This shows that the dipole is closely aligned to the long axis of the peptide. **d** Net charge of the peptide across a range of pH values. All residues in the charged region are charged between approximately pH 5–10, leading to a net charge of −4 e. **e** Space-filling model of a representative structure of the FLAG-tag peptide, showing the turn across residues F2–A7, and the extended random coil structure of the polar, charged residues (D16–K21). Scale bar: 1 nm

The detection of sulfur even after Tween 20 or sodium dodecyl sulfate (SDS) washing indicates that the peptide is attached covalently to the surface. The slight loss of signal is likely due to the removal of additional adsorbed peptide. SDS and Tween 20 are detergents that disrupt physical interactions between adsorbed solutes and surfaces while leaving covalent bonds intact.

The ELISA results for the RFPP surface functionalized with peptide show an increase in signal over the control sample for both pH values of 5.4 and 9.8, indicating the presence of peptide on the surface (Fig. 4d). The high concentration used here is saturated for the pH 5.4 condition, and since the pH 9.8 condition shows the same absorbance signal, it is also likely to be saturated. Despite the utility of ELISA for detecting low concentrations of surface peptide, the size differential between the peptide and the antibody molecules means that once bound, antibody molecules block access to other peptide molecules underneath the antibody footprint. This indicates the assay saturates at concentrations far below a peptide monolayer. Such behaviour is an intrinsic limitation of two-dimensional ELISA, particularly for peptides, which have a much smaller footprint than antibody proteins and so saturate at low coverage densities.

Although XPS and ELISA indicated the surface concentration of peptide, they could provide only limited information about its orientation. The orientation of peptides has been deduced before using highly surface-sensitive techniques such as sum frequency generation spectroscopy[59] and time of flight secondary ion mass spectroscopy (ToF-SIMS)[60]. In this study, we used ToF-SIMS as a highly surface-sensitive technique with a sampling depth of 1–2 nm[61]. This technique enables an assessment of peptide orientation by revealing amino acids dominant in the top-most region of the peptide layer. ToF-SIMS is performed in vacuum; therefore, the conformation of the peptide may be different compared to that in the aqueous environment. Nevertheless, the covalent linkage will minimize the sputtering of amino acids from the tethered end. To assess the orientation, we used the average positive SIMS counts associated with methionine and phenylalanine, normalized to the total 42 protein-associated secondary ion fragments[62] (Fig. 4e). The other amino acid residues from the peptide (alanine, lysine and aspartic acid) are excluded, because they cannot be unambiguously discriminated from either fragments originating from the RFPP layer or from each other (See Supplementary Fig. 2). Both residues show greater counts for the peptide immobilized at pH 9.8 compared to that immobilized at pH 5.4. These two amino acids make up the hydrophobic end of the peptide (Fig. 3e), indicating that there is a greater number of molecules in the FLAG_DOWN orientation for pH 9.8 than for pH 5.4. The difference in counts can be due to either a change in overall density of the peptide on the surface or in the preferred orientation of the immobilized peptides. XPS data, however, showed that the pH 5.4 condition yields a substantially higher overall peptide density. It can, therefore, be concluded that there is a strong preference for FLAG_DOWN orientation for peptide immobilized at pH 9.8 relative to that immobilized at pH 5.4.

XPS, ELISA and ToF-SIMS results together show the effect of immobilization pH on both the surface concentration and the orientation of bound peptide. At pH 5.4, where the surface charge is close to neutral, the repulsion between the RFPP and the negatively charged peptide is negligible, leading to higher

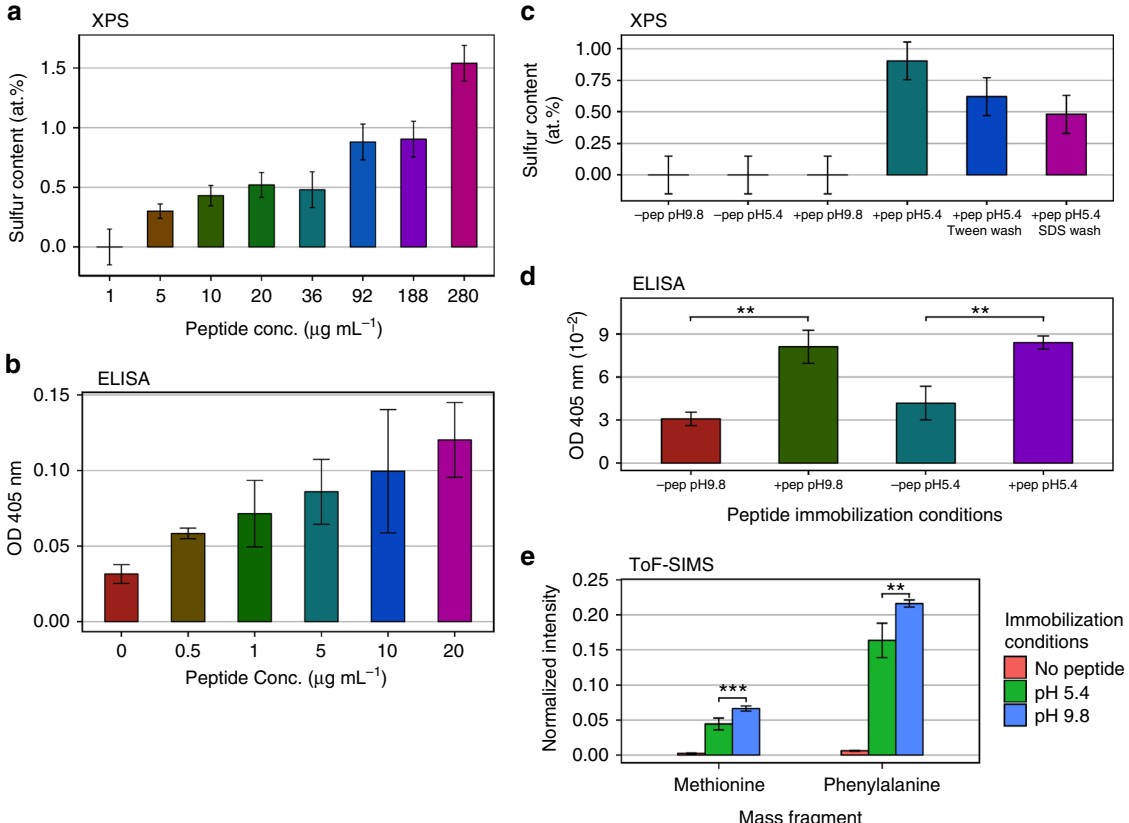

**Fig. 4** pH controls the peptide concentration and orientation. **a** Sulfur content, as measured by X-ray photoelectron spectroscopy (XPS), indicates increasing surface peptide with solution concentration but is undetectable for concentrations below 5 µg mL⁻¹. **b** Peptide titration shows that enzyme-linked immunosorbent assay (ELISA) is discriminatory for peptide concentrations down to at least 0.5 µg mL⁻¹ but saturates at approximately 20 µg mL⁻¹ due to the large footprint of the primary and secondary antibodies compared to the peptide. Hence, ELISA is not capable of discerning immobilized peptide surface density for high densities while XPS is not capable of detecting low peptide surface concentrations. **c** XPS sulfur atomic concentration of peptide-coated surfaces compared to uncoated controls for immobilization pH values of 5.4 and 9.8. The sulfur content shows surface peptide at pH 5.4, but no detectable peptide at pH 9.8. Presence of peptide after Tween 20 or sodium dodecyl sulfate (SDS) washing indicates covalent attachment. **d** The absorbance measured with ELISA for peptide-coated samples for immobilization pH values of 5.4 and 9.8 compared with uncoated controls. ELISA shows that peptide is present at both pH values. **e** Time of flight secondary ion mass spectrometry (ToF-SIMS) normalized mass fragments of the hydrophobic residues indicate higher number of FLAG$_{DOWN}$ peptide for pH 9.8, suggesting that a greater proportion of peptide for pH = 5.4 is in the FLAG$^{UP}$ orientation. Error bars are s.d. and $P$-values are from the Student's two-tailed $t$-test: **$P < 0.01$, ***$P < 0.001$ (**b**, **d** and **e**). XPS error bars are calculated from the background noise (**a**, **c**)

functionalization density. Because of the negligible charge interaction with the surface, the orientation is most likely random. Given that the dispersive component of the RFPP surface energy is higher than the polar component, there may be a preference for FLAG$^{UP}$ orientation due to hydrophobic interactions between the surface and the hydrophobic end of the peptide. In contrast, at pH = 9.8 the RFPP is more negatively charged, repelling the peptide and reducing its density on the surface by reducing the arrival rate. Under such conditions, the peptide is also preferentially oriented on the surface in the FLAG$_{DOWN}$ configuration. This orientation is favoured, because it aligns the dipole in the FLAG epitope created by the C-terminal aspartate and lysine residues with the electric field at the surface. These findings demonstrate that pH can be utilized to control surface orientation and concentration of immobilized peptide. The mechanism for this control is the appearance of an electric field normal to the surface when the surface is charged. This effect would increase as the surface charge increases, because the electric field is proportional to the surface charge density. For extra control in situations where the pH is constrained, the application of an external electric field could be an efficient alternative.

**Orientation and concentration control by external E-field.** Surface charge created by changes in solution pH is often limited by the types of chemical groups present on the surface. In situations where linker chemistry is used to immobilize biomolecules, pH ranges are often restricted to those that facilitate the required chemistry. Here we show that an externally applied electric field will overcome these limitations, enabling greater control of orientation and concentration. Specifically, the RFPP surface in this experiment is not able to achieve a significant positive charge using solution pH and hence is unable to attract the negatively charged peptide. We applied external electric fields in both directions during the peptide immobilization using a custom-made well-plate and a simple power supply. We denote the electric field direction using $E^{up}$ and $E_{down}$, corresponding to the positive terminal connecting to the lower or upper plate, respectively. The negative terminal was connected to the opposite plate in each case. The $E^{up}$ field is expected to attract additional peptide molecules to the surface leading to higher surface density. We used initially a low peptide concentration (1 µg mL⁻¹) in the immobilizing solution to demonstrate the enhancing effect of electric fields on surface density, while a high peptide concentration (180 µg mL⁻¹) was used to increase the signal-to-noise

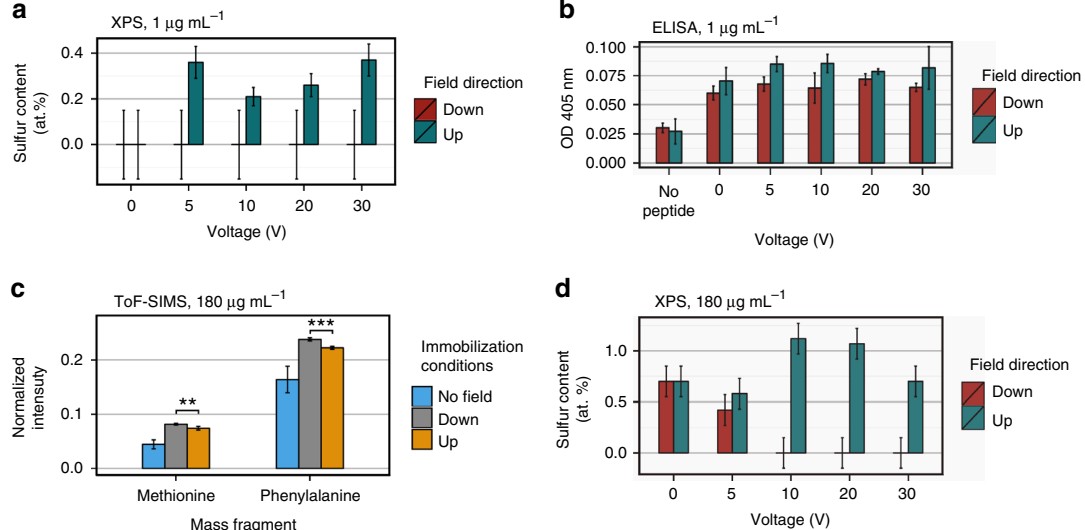

**Fig. 5** External field controls peptide density and orientation. **a** The sulfur atomic concentration of the peptide-functionalized surface from X-ray photoelectron spectroscopy (XPS) as a function of applied voltage (peptide solution concentration = 1 μg mL$^{-1}$). An increase in peptide arrival rate is observed for the $E^{up}$ field, while no peptide is detectable for the $E_{down}$ field. **b** The absorbance measured with enzyme-linked immunosorbent assay (ELISA) as a function of applied voltage showing that peptide is present after immobilization in both $E^{up}$ and $E_{down}$ field directions (peptide solution concentration = 1 μg mL$^{-1}$). **c** Time of flight secondary ion mass spectrometry (ToF-SIMS) normalized mass fragments of the hydrophobic peptide residues indicate approximately equal amounts of FLAG$_{DOWN}$ peptide (peptide solution concentration = 180 μg mL$^{-1}$). **d** The sulfur atomic concentration of the peptide-functionalized surface from XPS as a function of applied voltage (peptide solution concentration = 180 μg mL$^{-1}$). Surface peptide is increased by $E^{up}$, and decreased by $E_{down}$ relative to the case when no field is applied. Given the higher concentration for the $E^{up}$ field but equal amount of hydrophobic residues (indicating FLAG$_{DOWN}$ peptide) in ToF-SIMS, there is a higher proportion of FLAG$^{UP}$ peptide for this field direction compared to the $E_{down}$ direction. Error bars are s.d. and P-values are from the Student's two-tailed t-test: **P < 0.01, ***P < 0.001 (**b**, **c**). XPS error bars are calculated from the background noise (**a**, **d**)

ratio for the evaluation of orientation by XPS and ToF-SIMS. In all of the following experiments, the pH of the immobilizing solution was kept at 5.4 to reduce the influence of charged surface moieties.

We used XPS, ELISA and ToF-SIMS to characterize surfaces prepared with an electric field applied either in the $E^{up}$ or $E_{down}$ directions (Fig. 5). It is observed that the low peptide concentration in the immobilizing solution leads to a surface concentration below the detection limit of XPS for the $E_{down}$ field, while for $E^{up}$ the peptide is detectable through XPS sulfur signals reaching to 0.2–0.35 at% (Fig. 5a). This level of functionalization is comparable to that achieved using solution concentrations 5- to 10-fold greater in the absence of an electric field (see Fig. 4a). Such improvement of peptide surface density suggests a great potential to use an external electric field during immobilization to increase the yield of bound peptide, which is particularly important for surface bio-functionalization using expensive molecules.

There is an ELISA signal above the no-peptide control at each voltage and field direction applied, indicating that peptide is present in all cases, even for $E_{down}$ where peptide was not detected by XPS (Fig. 5b). Unlike the XPS signal, the ELISA absorbance does not significantly vary across the range of applied voltages, indicating that the signal is saturated. The difference between the saturated values for the $E^{up}$ and $E_{down}$ conditions is statistically significant ($P < 0.001$). This possibly arises due to suboptimal packing of antibody on the lower density surface ($E_{down}$) caused by greater gaps without peptide on the surface where antibodies cannot bind. On the $E^{up}$ surface, the higher density of peptide facilitates closer packing of the larger antibodies and therefore a slightly higher saturated signal.

To investigate differences in orientation, we employed XPS and ToF-SIMS. As discussed before, only the phenylalanine and

methionine amino acid residues from the hydrophobic region of the peptide are analysed to infer orientation by ToF-SIMS data. Both electric field directions show higher concentrations of methionine and phenylalanine on the topmost layer compared to the sample prepared in the absence of electric fields (Fig. 5c). This is indicative of greater concentration of FLAG$_{down}$ oriented peptide in both cases. The two field directions show the same relative amounts, with a slightly higher signal for each residue for $E_{down}$ than $E^{up}$.

ToF-SIMS results can be better understood by taking into account the XPS results for samples prepared with and without the application of external electric fields (Fig. 5d). The sulfur atomic concentration is increased for the $E^{up}$ field and decreased for the $E_{down}$ field, indicating that the peptide concentration for the $E_{down}$ field reduces as the electrical potential is increased. Similar to the pH 9.8 condition, the $E_{down}$ field creates a potential barrier for the negatively charged peptide as it approaches the surface and hence repels the peptide molecules. In contrast, when the field is in the upward direction ($E^{up}$), a potential well for negatively charged peptide is created at the surface, attracting greater numbers of peptide molecules to the surface. This situation did not arise in the pH comparison experiment since we were limited to a negative surface charge in order to have a fully charged peptide. The application of the $E^{up}$ field thus demonstrates the extra control possible over functionalization. This is particularly useful given the importance of optimizing the surface concentration of bioactive peptides[24,26,35,63,64].

Using the information about the density of surface peptide in each case, as well as the concentration of FLAG$_{DOWN}$-oriented peptide molecules given by ToF-SIMS, we can now infer the concentration of FLAG$^{UP}$-oriented peptide. According to XPS results, a higher density of surface peptide is achieved by applying the $E^{up}$ field compared to the $E^{down}$ configuration. ToF-SIMS

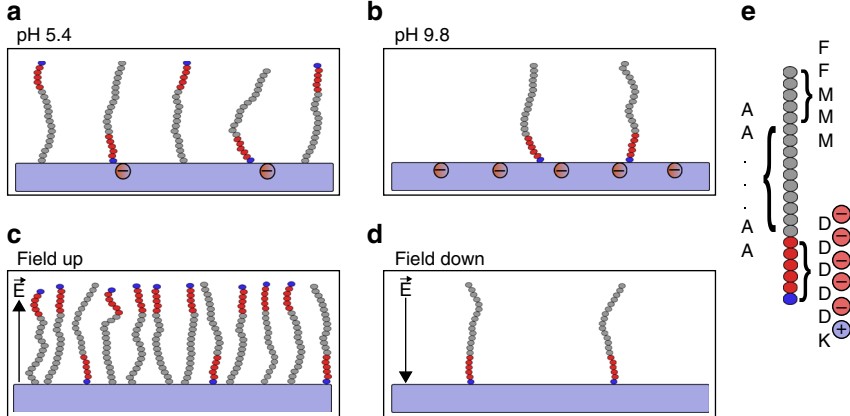

**Fig. 6** Illustration of peptide orientation and concentration. The immobilization conditions were **a** pH 5.4, **b** pH 9.8, and with applied electric fields **c** $E^{up}$ and **d** $E_{down}$ both at pH 5.4. The radical-functionalized plasma polymer surface is indicated by purple rectangles, with charges on both the surfaces and the peptide indicated by circled + or – signs. **e** The enlarged peptide indicates the position of amino acids (lysine (K) blue, aspartic acid (D) red, and all others shown in grey)

data, however, suggested that the concentration of FLAG$_{DOWN}$ molecules are approximately equal in both cases. It can, therefore, be concluded that the majority of peptide molecules immobilized using the $E^{up}$ field are in a FLAG$^{UP}$ orientation. In contrast, for the $E_{down}$ field there is mostly FLAG$_{DOWN}$ peptide. This distinction between the surfaces balances out such that the total presentation of hydrophobic residues is approximately equal, as shown by ToF-SIMS. We have therefore demonstrated that stark differences in orientation and concentration of peptide molecules are achieved through the application of external electric fields during the immobilization process. Importantly, unlike other recent studies where orientation is used to optimize peptide activity[28,31,34,52,53,65], such differences are achieved without adding any extra amino acids for covalent linkage or changing the immobilizing solution.

## Discussion
Here we discuss our findings for the variations of concentration and orientation as a function of electric field changes induced by pH or potential differences applied across the immobilizing solution. For controlling the peptide functionalization by modulating the immobilizing solution pH, higher surface densities were achieved when the surface was less charged. Since the surface charge is small, and so has a negligible electrostatic effect on the peptide, the arrival rate and subsequent covalent linkage are mediated by the hydrophobic effect and van der Waals interactions only. In this case, the peptide binds in a random orientation (Fig. 6a). When the surface charge is more negative at higher pH values (Fig. 6b), the total peptide density is lower due to the potential barrier created by the increased negative surface charge. However, when peptide molecules do bind to the surface, they are oriented FLAG$_{DOWN}$ in order to align their dipole moment with the electric field created by the charged surface. An externally applied electric field is also capable of influencing the peptide-surface interactions. For the $E_{down}$ field (Fig. 6d), the negative bias on the surface repels peptide molecules, reducing the density of peptides immobilized on the surface and simultaneously orienting them FLAG$_{DOWN}$ (as for immobilization at pH 9.8). For the $E^{up}$ field (Fig. 6c), which was not achievable by pH modulation, the surface is positively biased and thus more peptide molecules are attracted, leading to a high immobilized surface density. Additionally, the field orients peptides in the FLAG$^{UP}$ orientation. For surface functionalization, this represents the best-case scenario where high concentration of bound peptide as well as an increased proportion of the correct orientation are achieved.

We believe that our findings on the mechanisms of peptide immobilization controlled by electric fields have important implications for the design of synthetic biomolecules and bio-functionalization of advanced implantable materials. In particular, guidelines for the design of synthetic peptides to optimize their presentation and density when immobilized can be inferred. For a particular functional peptide sequence and radical-functionalized surface, the amino acid charge sequence and surface potential as a function of pH must be determined so that an optimal pH for immobilization can be identified. If there is no pH at which the peptide net charge and dipole moment are favourable, then additional amino acids can be appended to the peptide end that is to be bound at the surface so as to adjust both the dipole moment and the charge. Alternatively, electric fields as shown in this work can be applied to achieve the optimal orientation and density. Our findings suggest that tuning the immobilization solution pH and/or the application of electric fields during immobilization have the potential to improve bio-functionalization methods, offering better outcomes for implantable devices used in modern biomedicine.

## Methods
**Materials**. Titanium substrates (thickness = 0.07 mm) were obtained from Firmetal, China, and were ultrasonicated in acetone and ethanol, rinsed with Milli-Q water, and dried using a nitrogen gas stream prior to RFPP film deposition. High purity argon, acetylene and nitrogen gases were supplied by BOC, Australia. Buffer reagents, Tween 20 and SDS were purchased from Sigma Aldrich. Goat primary polyclonal anti-DDDDK antibody and rabbit anti-goat secondary antibody were obtained from Abcam, Australia. The designed peptide (Ac-FFMMMAAAAAAAAAAADDDDDK-NH$_2$) was purchased from Auspep at 95% purity.

**Plasma polymerization**. The deposition of a thin RFPP layer on titanium substrates was performed using a custom-made plasma polymerization system, described elsewhere in detail[46,66]. A precursor gas mixture of argon, acetylene and nitrogen was injected into the chamber at flow rates of 15, 5 and 10 standard cubic centimetres per minute (sccm), respectively. The system working pressure was kept constant at 110 mTorr, while the base pressure was below $5 \times 10^{-2}$ mTorr. Plasma polymerization was performed for 2 min at an RF input power of 50 W provided by an ENI radio frequency power generator (13.56 MHz). Voltage pulses of $-500$ V (pulse duration = 20 µs, frequency = 3 kHz) were applied to the substrate holder using an RUP-6 pulse generator (GBS Elektronik GmbH). Titanium samples were cleaned by argon plasma prior to plasma polymerization as the final cleaning step. To achieve a stable surface chemistry, the PP-coated samples were stored in the laboratory environment for 7 days prior to incubation with the peptide.

**X-ray photoelectron spectroscopy (XPS)**. Surface chemistry of PP-coated surfaces before and after peptide immobilization was analysed using a SPECS FlexMod spectrometer. The instrument was equipped with a monochromatic Al K$_\alpha$ ($h\vartheta =$

1486.7 eV) radiation source operating at 200 W. A PHOIBOS 150 hemispherical analyser and an MCD9 electron detector were used for spectroscopy. The electron take-off angle was 90° referenced to the sample surface, and the base pressure was always maintained below $5 \times 10^{-8}$ mbar. Survey spectra were recorded in an energy range of 0–1000 eV with a resolution of 0.5 eV (pass energy = 30 eV). High resolution C 1s, S 2p, N 1s, and O 1s spectra were collected at a resolution of 0.1 eV and a pass energy of 20 eV. The high-resolution spectra were used for surface chemical composition calculations. Processing and atomic concentration calculations were performed using CasaXPS software (version 2.3.1).

**Contact angle measurements**. The contact angles of the RFPP surface were measured using a Krüss DSA Mk2 goniometer. Eight measurements were taken for both water and diiodomethane using the sessile drop method by applying 1 μL drops. The surface energy was calculated using the Owens–Wendt model[67], which relates the contact angle and polar and dispersive components of the liquids to the surface energy.

**Electron paramagnetic resonance (EPR) spectroscopy**. EPR spectroscopy was conducted using a Bruker EMXplus Xband to evaluate the radical functionalization of the RFPP coating. RFPP-coated polystyrene films (7 cm×7 cm) were rolled into Wilmad Borosilicate glass NMR tubes, and the spectrometer was calibrated using a weak pitch sample. Spectra were recorded with a central magnetic field of 3510 G, modulation amplitude of 3 G, microwave frequency of 9.8 GHz and power of 25 mW. The field modulation frequency was $1 \times 10^5$ and the sampling time was 85 ms. Ten scans were averaged per sample.

**Electro kinetic analysis**. The zeta potential of the RFPP surface was measured as a function of pH using an Anton Paar SurPASS Electrokinetic Analyzer with an automatic titration unit to adjust the pH between 3 and 10. Two 10×20 mm

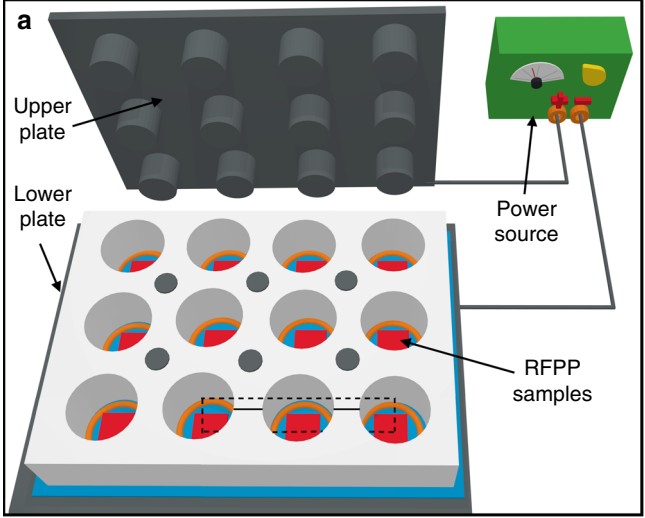

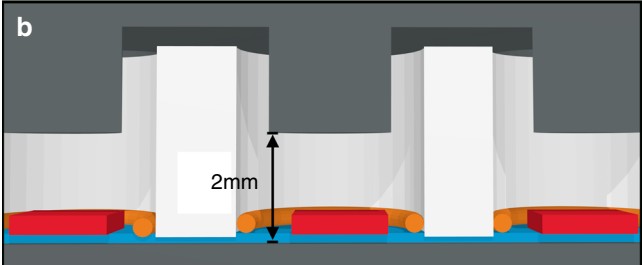

**Fig. 7** The sample holder for applying external electric fields. **a** The radical-functionalized plasma polymer-coated samples (red) lie in wells in a polyether ether ketone sample holder with 12 wells (white). Rubber O-rings (orange) seal each well, which are insulated by a 0.1 mm polytetrafluoroethylene sheet (blue). The upper plate fits on top of the sample holder, and electrical potential is applied across the wells using a variable power source. Dotted lines indicate the cross-section in (**b**). **b** Cross-section of three wells. The extensions from the top plate lower into each well, leaving a 2-mm gap between the bottom and top plates. The sample is raised 0.1 mm above this level by the insulating material

samples of RFPP were fixed to an adjustable cell sample holder with a gap of approximately 100 μm between the samples. The measurement solution of $1 \times 10^{-3}$ M KCl was adjusted to an initial pH of 10 using appropriate volumes of 1 M NaOH. At each pH value, this solution was pumped through the sample holder, in both directions, with a pressure ramp to measure the streaming current. Zeta potential was then calculated using the Helmholtz-Smoluchowski equation. Four measurements were taken at each pH, before adjustment by aliquots of HCl by the automatic titration unit. The reported zeta potential values are averages at each pH value.

**Peptide simulation**. Two molecular dynamics equilibrium simulations were run to assess the secondary-structural behaviour of the FLAG peptide, each using different starting structures. The peptide starting coordinates were generated using Avogadro[68] software as either fully linear or alpha-helix. These structures were solvated with TIP3P water using the VMD Solvate plugin[69], with box sizes 14 Angstrom larger than the peptide's longest axis in the $x$, $y$ and $z$ directions using periodic boundary conditions. This ensured the peptides did not interact with their periodic images in an adjacent box. Each system had 150 mM NaCl added. The systems were minimized for 4000 steps and the water equilibrated for 1 ns by restraining the alpha-carbon atoms.

The alpha helical peptide equilibrium simulation was run for 300 ns. The alpha helix conformation was unstable and by this point had unfolded. This simulation was stopped since the peptide was longer and could interact with periodic copies of itself. The linear peptide simulation was run for 800 ns. Both simulations used a constant pressure of 1 atm and temperature of 298 K maintained using a Langevin Piston and Langevin thermostat respectively. The CHARMM27 force field[70] was used for the protein parameters, and the simulations, using a 2 fs timestep, were run using NAMD2.91.

**Peptide immobilization**. The buffer used for immobilization was designed to have equal ionic strength and buffer capacity at a low and a high pH value. A mixed buffer system was therefore used, with two buffers with different pKa values, allowing similar buffering capacity at two pH values. The buffer system consisted of an acetate/acetic acid buffer (pKa 4.8) and an ammonia/ammonium buffer (pKa 9.2) in the same solution. The pH was increased by 0.6 from the pKa to ensure that the peptide would fully dissolve at the lower pH value while maintaining its maximum charge. At pH = 5.4 the acetate/acetic acid buffer predominates, whereas at pH = 9.8 the ammonia/ammonium buffer predominates, while the buffering capacity is equal in both cases. The buffers were prepared by adjusting the pH of an ammonium acetate solution using either concentrated HCl or NaOH and adding NaCl to equalize the ionic strength.

The FLAG peptide was dissolved in 10 mM buffer at either pH 5.4 or pH 9.8. The incubation on the plasma polymer surface was performed by pipetting 500 μL of peptide solution onto a 1 cm×1 cm sample in a 24-well plate, when no external electric field was being applied. For samples with applied external electric field, the incubation was performed using a custom-made apparatus, schematically shown in Fig. 7. This apparatus consisted of wells of the same dimensions as the 24-well plate, but with electrodes above and below the sample. Each sample was incubated for 20 h at room temperature with or without an electric field applied. A polytetrafluoroethylene foil was placed over the bottom electrode, which ensured that no current was drawn. After incubation, the peptide solution was removed, and the sample was washed three times using 1 mL de-ionized water. To validate the covalent attachment of the peptide, the detergent-washed samples were placed in a 5% Tween 20 or 5% SDS solution heated to 70 °C for 1 h, then washed three times using 1 mL de-ionized water.

**Time of flight secondary ion mass spectrometry (ToF-SIMS)**. To evaluate the orientation of the immobilized peptide on the RFPP-coated surfaces, ToF-SIMS measurements were performed using a Physical Electronics Inc. PHI TRIFT V nanoTOF instrument. A pulsed primary $^{79}Au^+$ ion beam (30 keV) was used for the ionization of species. Dual charge neutralization was achieved using a PHI system by applying a combination of low energy argon ions (10 eV) and electrons (25 eV). The base pressure was $5 \times 10^{-4}$ Pa or less during the measurements. The spectra were collected in bunched instrument settings to optimize mass resolution. Positive SIMS data were acquired over areas of 100 μm×100 μm for an acquisition time of 60 s. For each sample, six spectra were recorded at different locations to assess repeatability. Species associated with protein secondary ion fragments[62] were selected, and the counts were normalized to the total intensity of all selected peaks. Acquired data were processed and integrated using WincadenceN software (Physical Electronics Inc. Chanhassen, MN, USA). Error bars for ToF-SIMS represent the 95% confidence interval calculated from the six recorded spectra.

**Enzyme-linked immunosorbent assay (ELISA)**. ELISA was performed to detect bound peptide on the RFPP surface. The peptide-functionalized samples were blocked with 3% BSA solution. Peptide functionalization was detected using the goat primary polyclonal anti-DDDDK antibody ab1257, followed by a rabbit anti-goat secondary antibody ab6741. The samples were added to ABTS solution and absorbance was measured at 405 nm after 1 h incubation. Statistical significance was calculated using single factor analysis of variance.

**Data availability**. The authors declare that all data supporting the findings of this study are available within the article and its supplementary information files or from the corresponding authors upon request.

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

## Acknowledgements

We acknowledge the Australian Research Council for funding this research and the National Computational Infrastructure (NCI), supported by the Australian Government, for the computational resources used for our peptide simulations. L.J.M. acknowledges scholarship top-up funding from the CRC for Cell Therapy Manufacturing. Many thanks to Edgar Wakelin for design of the sample holder. We thank Dr. Clare Hawkins of the Heart Research Institute for access to and assistance with EPR and Professor Anthony Weiss and Dr. Giselle Yeo for access to and assistance with ELISA. Thanks also to Shao Dai and Ahmad Jabbarzadeh of the School of Aerospace, Mechanical and Mechatronic Engineering for access to and assistance with zeta-potential measurements. We acknowledge the facilities of the Australian Microscopy & Microanalysis Research Facility. We are thankful to Dr. John Denman for undertaking ToF-SIMS measurements and to Professors Richard Payne and David McKenzie and Associate Professor Boris Kuhlmey for providing helpful comments on the manuscript.

## Author contributions

L.J.M. and B.A. contributed equally to this work. M.M.M.B. proposed the concept. L.J.M. and B.A. performed the experiments. L.J.M., B.A. and M.M.M.B. analysed the results and wrote the manuscript.

## Additional information

**Competing interests:** The authors declare no competing financial interests.

