## [Peer Review File · Nature Communications]

Reviewers' comments:

Reviewer #1 (Remarks to the Author):

>What are the major claims of the paper?

The authors present a method for controlling the orientation of peptides during immobilization on a solid-substrate. The authors used electric fields to control the peptide approach towards a radical functionalized surface. The findings show that control of peptide orientation and surface concentration can be varied with electric field strength.

>Are they novel and will they be of interest to others in the >community and the wider field?

The claims in this paper are not novel. Immobilization of macromolecules with controlled orientation was first demonstrated by Emaminejad et. al (PNAS 2015), and this paper makes no mention of that. The immobilization of peptides with controlled orientation was also first presented by Price and Utz and colleagues in 2012 (not using fields). Thus, this paper is not at the novelty required for Nature Communications. No mention was made of either of these works in the introduction, thus the authors should do a more comprehensive literature search before making such bold claims about the novelty of their work.

Although this work is meritorious and should be published somewhere, it belongs in a more specialized journal, rather than Nature communications, which is more for a broad audience and very novel original work.

>If the conclusions are not original, it would be helpful if >you could provide relevant references. Is the work >convincing, and if not, what further evidence would be >required to strengthen the conclusions?

The only way this would be at the caliber of a journal like Nature Communications would be for the authors to back up their claims about biofouling mitigation in-vivo. This would require testing the device in-vivo and to validate that the use of electric fields does indeed lower fouling of the biomaterial surface.

All of the data they show is in purified and very controlled conditions.

>On a more subjective note, do you feel that the paper will >influence thinking in the field?

I think the references mentioned already do so. This is incremental progress (not transformative) over the existing work in the literature.

Having said this, I recommend the editorial board to reject this manuscript and encourage the authors to submit this work to a more specialized journal. The only thing that could make this work suitable for NC, is to characterize biofouling in an in-vivo setting.

Reviewer #2 (Remarks to the Author):

The manuscript "Electric fields control the orientation of peptides irreversibly immobilized on radical-functionalized surfaces" by Bilek et al. is structured and written well and clearly outlines a novel concept for oriented surface immobilization. I suggest that the paper is accepted after the

following issues have been addressed by the authors:

The authors demonstrate a reasonable difficult concept using a range of analytical techniques. To make it easier for the reader to comprehend the data, the authors should add further labels to figures. An example is Figure 6a-d, where labels should be added to each of the 4 sections to show the different protein concentrations used and the different analytical techniques used.

The authors should clarify where in the peptide the sulfur is found - Figure 4 is an opportunity for this. Again this will make it easier for the reader in regard to the interpretation of XPS data.

Throughout the manuscript, the authors should make it clear by adding further labelling and definitions which of the data are statistically significant and which are not.

The authors should provide additional data if available to clarify if the surface immobilization has exclusively occurred covalently or if there are other contributing factors.

Reviewer #3 (Remarks to the Author):

This paper reported a method to control the surface immobilized peptide coverage and orientation. The peptides were immobilized on a radical functionalized polymer surface. The peptide coverage and orientation can be varied using different pH values of the solution or an external electric field. Computer simulations were performed to predict the peptide structure after surface adsorption. XPS, ELISA, and ToF-SIMS were used to investigate the peptide structure on surface. The results are interesting and should be accepted for publication after minor revisions:

(1) Most medical devices or biomedical materials are used in aqueous environments, therefore it is important to understand the structures of biological molecule coatings in aqueous environments. The surface immobilized peptides studied in this research were prepared in aqueous condition as well. However, the immobilized peptides studied by XPS and ToF-SIMS were in high vacuum. Surface immobilized peptides can have markedly different orientations in different chemical environments, e.g. aqueous solution vs. vacuum. I suggest the authors to add a statement to the discussion section to emphasize that the results obtained from the XPS and ToF-SIMA studies are from surface immobilized peptides in vacuum, they may not be the same as those obtained in an aqueous solution.

(2) The authors qualitatively deduced the absolute orientation of the peptides immobilized on surface (up or down). Using polarized optical spectroscopic method (e.g., sum frequency generation vibrational spectroscopy and/or attenuated total reflection FTIR), it is feasible to quantify the orientation angle of surface immobilized peptides at the solid/liquid interface in situ. The authors should mention this in the discussion section as well.

We thank you and the reviewers for careful reading of our manuscript entitled “Electric fields control the orientation of peptides irreversibly immobilized on radical-functionalized surfaces” (NCOMMS-17-19718) and for the constructive comments. We have addressed all the comments and incorporated the suggestions into the revised manuscript.

We provide here the reviewer’s comments, and our response in each case detailing how each comment was addressed. We believe these revisions have improved the manuscript.

Reviewers’ comments to authors are in bold with authors’ responses in italics:

Reviewer 1.

>What are the major claims of the paper?

The authors present a method for controlling the orientation of peptides during immobilization on a solid-substrate. The authors used electric fields to control the peptide approach towards a radical functionalized surface. The findings show that control of peptide orientation and surface concentration can be varied with electric field strength.

>Are they novel and will they be of interest to others in the community and the wider field? The claims in this paper are not novel. Immobilization of macromolecules with controlled orientation was first demonstrated by Emaminejad et. al (PNAS 2015), and this paper makes no mention of that. The immobilization of peptides with controlled orientation was also first presented by Price and Utz and colleagues in 2012 (not using fields). Thus, this paper is not at the novelty required for Nature Communications. No mention was made of either of these works in the introduction, thus the authors should do a more comprehensive literature search before making such bold claims about the novelty of their work.

We thank the reviewer for bringing these references to our attention. We now include reference to them and extend our introduction to clarify the novelty of our approach. Our approach is distinct from both of these two approaches and provides significant advantages as described below:

The article in PNAS examines dielectric response of a large protein (~150 kDa), whilst our work examines the behaviour of small peptides (~2 kDa). The fundamental mechanism responsible for the occurrence of alignment is completely different due to this discrepancy in size. For large molecules (>100 kDa [1,2]), the induced dipole moment dominates the permanent dipole moment, and hence only bidirectional alignment along the long axis can be achieved. We rely on the dominance of the permanent dipole moment for small peptides to achieve unidirectional orientation normal to the surface.

Our experimental approach is also distinct and advantageous. The approach described in the PNAS article relies on faradaic electrodes to sustain a field across the bulk solution. Faradaic electrodes modify the chemistry of the solution through electrochemical reactions and degrade with time in use. In contrast, our approach is implemented with inert electrodes, exploiting the electric fields in the double layer to achieve control of orientation. Inert electrodes allow the field to be applied externally avoiding the need to fabricate electrodes into the vessel containing the solution and hence dramatically simplifying the process.

The article by Price and Utz describes the use of chemical methods that rely on specific functional groups being available on both the peptide and the surface. Chemical methods of orienting biomolecules are well known; but the sequence of wet chemical reactions required makes the process cumbersome and prone to leaving potentially toxic residuals. Our approach does not require any specific chemical groups to be present or rely on specific wet-chemistry.

To address the reviewer’s concern, we have added the new references and extended the Introduction Section with the following text:

Page 4: “Control of the surface orientation of peptides by changing their chemistry and achieving site-specific attachment has been studied before. For example, site-specific phosphorylation of serine residues in a silicon based peptide³⁶, or the insertion of a cysteine residue³⁷. However, controlling the orientation of peptides with these traditional chemical approaches is typically cumbersome and tedious for large-scale manufacturing.”

Page 4: "The electric field at the surface can be manipulated by changing the solution pH or by the application of an external electric field. Electric field induced bidirectional alignment along the long axis of large proteins has been achieved with DC fields³⁸ and in dielectrophoresis with AC electric fields³⁹. A unidirectional orientation can only be achieved via a permanent dipole moment and hence requires small molecules (< ~100 kDa), for which the induced dipole moment does not dominate⁴⁰."

[1] Takashima, Shiro. "Dielectric dispersion of protein solutions in viscous solvent." *Journal of Polymer Science Part A: Polymer Chemistry* 56.163 (1962): 257-265.

[2] Leach, Sydney, ed. *Physical principles and techniques of protein chemistry*. Elsevier (2012): pg 321-322

Although this work is meritorious and should be published somewhere, it belongs in a more specialized journal, rather than Nature communications, which is more for a broad audience and very novel original work.

As described above the work is novel and clearly distinct from the cited references. We also believe that it is intrinsically of interest to a broad audience. The use of small synthetic peptides that can recapitulate particular functions of native biological molecules is rapidly gaining momentum. In the context of practical devices, peptides offer the advantages of simple, cost-effective synthesis compatible with GMP manufacturing and enhanced stability. In order to maximise peptide activity when immobilised, control of orientation is necessary. The substantial and rapidly growing interest in using peptides across many fields and applications ensures that this research is of interest to a broad audience.

>If the conclusions are not original, it would be helpful if you could provide relevant references. Is the work >convincing, and if not, what further evidence would be required to strengthen the conclusions?

The only way this would be at the caliber of a journal like Nature Communications would be for the authors to back up their claims about biofouling mitigation in-vivo. This would require testing the device in-vivo and to validate that the use of electric fields does indeed lower fouling of the biomaterial surface.

All of the data they show is in purified and very controlled conditions.

As explained above, the simple approach to orient peptides described in our manuscript is original. In vivo studies are outside the scope of this article and will be pursued in subsequent works.

>On a more subjective note, do you feel that the paper will >influence thinking in the field? I think the references mentioned already do so.

This is incremental progress (not transformative) over the existing work in the literature. Having said this, I recommend the editorial board to reject this manuscript and encourage the authors to submit this work to a more specialized journal. The only thing that could make this work suitable for NC, is to characterize biofouling in an in-vivo setting.

We believe that the work is transformative because it provides for the first time a simple and practical approach to control the orientation of small peptides. Biofouling is just one of the many applications that our work is relevant to and as such it is best documented in a separate application-specific manuscript.

Reviewer 2.

The manuscript "Electric fields control the orientation of peptides irreversibly immobilized on radical-functionalized surfaces" by Bilek et al. is structured and written well and clearly outlines a novel concept for oriented surface immobilization. I suggest that the paper is accepted after the following issues have been addressed by the authors:

The authors demonstrate a reasonable difficult concept using a range of analytical techniques. To make it easier for the reader to comprehend the data, the authors should add further labels to figures. An example is Figure 6a-d, where labels should be added to each of the 4 sections to show the different protein concentrations used and the different analytical techniques used.

For the reader's clarification, we have added labels that describe the source of the data in each panel for the figures where data from multiple analytical techniques are presented, i.e. Figures 4 and 5 (Figures 5 and 6 in the originally submitted manuscript).

The authors should clarify where in the peptide the sulfur is found - Figure 4 is an opportunity for this. Again this will make it easier for the reader in regard to the interpretation of XPS data.

We have added a label on Figure 3 (Figure 4 in the originally submitted manuscript) indicating the amino acids that contain sulfur.

Throughout the manuscript, the authors should make it clear by adding further labelling and definitions which of the data are statistically significant and which are not.

Statistical significance labels are now included on figures where a direct comparison is made between two conditions, namely Figs. 4d, 4e, and 5c.

The authors should provide additional data if available to clarify if the surface immobilization has exclusively occurred covalently or if there are other contributing factors.

To confirm the covalent immobilization of the peptide molecules on the surface, the peptide-immobilized samples were analysed using XPS after washing with Tween 20 detergent heated to 70° C for 1 hr. Tween 20 is an ionic surfactant that removes physisorbed molecules on the surface but leaves behind those covalently attached to the surface. The XPS results showed minimal reduction of sulfur atomic concentration upon Tween 20 wash indicating that the majority of peptide molecules are attached covalently to the RFPP surface. These data were already presented and discussed in the paper (Figure 4).

To further confirm the covalent nature of the peptide-RFPP coupling, we have included the XPS atomic concentration of sulfur after washing with sodium dodecyl sulfate (SDS) as well. These results have been added to Figure 4. Text on Page 7 has been revised accordingly:

"The detection of sulfur even after Tween 20 or sodium dodecyl sulfate (SDS) washing indicates that the peptide molecules are attached covalently to the surface. The slight loss of signal is likely due to the loss of adsorbed peptide. SDS and Tween 20 are detergents that disrupt physical interactions between adsorbed solutes and surfaces while leaving covalent bonds intact."

Reviewer 3.

This paper reported a method to control the surface immobilized peptide coverage and orientation. The peptides were immobilized on a radical functionalized polymer surface. The peptide coverage and orientation can be varied using different pH values of the solution or an external electric field. Computer simulations were performed to predict the peptide structure after surface adsorption. XPS, ELISA, and ToF-SIMS were used to investigate the peptide structure on surface. The results are interesting and should be accepted for publication after minor revisions:

(1) Most medical devices or biomedical materials are used in aqueous environments, therefore it is important to understand the structures of biological molecule coatings in aqueous environments. The surface immobilized peptides studied in this research were prepared in aqueous condition as well. However, the immobilized peptides studied by XPS and ToF-SIMS were in high vacuum. Surface immobilized peptides can have markedly different orientations in different chemical environments, e.g. aqueous solution vs. vacuum. I suggest the authors to add a statement to the discussion section to emphasize that the results obtained from the XPS and ToF-SIMS studies are from surface immobilized peptides in vacuum, they may not be the same as those obtained in an aqueous solution.

We accept that XPS and ToF-SIMS are performed in high vacuum, which would compress peptides towards the surface; however, we don't believe this will change the conclusions regarding the peptide orientations. For XPS, which was used to measure the sulfur atomic concentration and thus measure peptide concentration, the sampling depth is 8 – 10 nm. This is greater than the height of even a fully extended peptide (~4.5 nm), and so we did not infer orientation from the XPS measurements.

ToF-SIMS was used in concert with XPS to determine the peptide orientation. The sampling depth of ToF-SIMS is 1-2 nm which is much smaller than that for XPS. Such high surface sensitivity of this technique allows analysing the very top most section of the peptide that is exposed on the surface. While the peptides were free to rotate in solution during the immobilization phase, after contacting the surface, covalent bonding to the surface “locks” the peptides in a distinct orientation. Even if the rest of the peptide is pushed towards the surface under vacuum, ultimately only one end is covalently bound, and this end will be closest to the surface. Thus, on average, the most accessible parts of the peptide (those measured by ToF-SIMS) will correspond to the orientation in the aqueous phase.

For further clarity, we added the following statement to the manuscript (Page 8):

“Although ToF-SIMS analysis is performed under vacuum, it is highly surface sensitive; so, the peptide orientation inferred in our experiments will reflect that in an aqueous environment. The covalent binding to the RFPP surface locks the peptides in a distinct orientation, and therefore the top most section of the peptide collapsed on the surface is expected to be similar in vacuum and aqueous environments.”

(2) The authors qualitatively deduced the absolute orientation of the peptides immobilized on surface (up or down). Using polarized optical spectroscopic method (e.g., sum frequency generation vibrational spectroscopy and/or attenuated total reflection FTIR), it is feasible to quantify the orientation angle of surface immobilized peptides at the solid/liquid interface in situ. The authors should mention this in the discussion section as well.

We initially used attenuated total reflection FTIR equipped with a germanium crystal to detect the peptides on the surface. Unfortunately, the technique was not surface-sensitive enough to detect a monolayer of peptides with reliable signal to noise ratios.

Regarding the surface-sensitive sum frequency generation spectroscopy, we have included the following statement in the manuscript (Page 8) to clarify other possible options:

“The orientation of peptides has been deduced before using highly surface-sensitive techniques such as sum frequency generation spectroscopy (SFGS)⁵⁹ and time of flight secondary ion mass spectroscopy (ToF-SIMS)⁶⁰. In this study, we used ToF-SIMS as a highly surface-sensitive technique with a sampling depth of 1 - 2 nm⁶¹.”

Reviewers' Comments:

Reviewer #1:

Remarks to the Author:

points have been satisfactorily addressed. I am ok with acceptance of the manuscript.

Reviewer #2:

Remarks to the Author:

The authors have fully addressed my suggestions and comments. I suggest that the manuscript is published without further changes.

Reviewer #3:

Remarks to the Author:

The author tried to address the comments I made for the last submission. I do not agree with the response to the first comment.

To address comment 1, the author added:

For further clarity, we added the following statement to the manuscript (Page 8): "Although ToF-SIMS analysis is performed under vacuum, it is highly surface sensitive; so, the peptide orientation inferred in our experiments will reflect that in an aqueous environment. The covalent binding to the RFPP surface locks the peptides in a distinct orientation, and therefore the top most section of the peptide collapsed on the surface is expected to be similar in vacuum and aqueous environments."

I do not agree with the statement "The covalent binding to the RFPP surface locks the peptides in a distinct orientation, and therefore the top most section of the peptide collapsed on the surface is expected to be similar in vacuum and aqueous environments". The peptides can behave completely differently in vacuum (e.g., linear, lying down) and in aqueous environments (e.g., standing up, bent). I suggest that the authors just mention that SIMS only gets data in vacuum. The real situation in aqueous solutions may or may not be similar.

We provide here the reviewer's comment, and our response.

Reviewer comment to authors is in bold with authors' response in italics:

Reviewer 3.

I do not agree with the statement "The covalent binding to the RFPP surface locks the peptides in a distinct orientation, and therefore the top most section of the peptide collapsed on the surface is expected to be similar in vacuum and aqueous environments". The peptides can behave completely differently in vacuum (e.g., linear, lying down) and in aqueous environments (e.g., standing up, bent). I suggest that the authors just mention that SIMS only gets data in vacuum. The real situation in aqueous solutions may or may not be similar.

We have replaced the statement with the following on page 8:

"ToF-SIMS is performed in vacuum; therefore, the conformation of the peptide may be different compared to that in the aqueous environment. Nevertheless, the covalent linkage will minimize the sputtering of amino acids from the tethered end."